# L$_1$-Norm Robust Regularized Extreme Learning Machine with Asymmetric C-Loss for Regression

**Qing Wu** [1,2] [iD], **Fan Wang** [1,*], **Yu An** [3] **and Ke Li** [1]

1   School of Automation, Xi'an University of Posts and Telecommunications, Xi'an 710121, China
2   Xi'an Key Laboratory of Advanced Control and Intelligent Process, Xi'an 710121, China
3   School of Electronic Engineering, Xi'an University of Posts and Telecommunications, Xi'an 710121, China
*   Correspondence: wangfan@stu.xupt.edu.cn

**Abstract:** Extreme learning machines (ELMs) have recently attracted significant attention due to their fast training speeds and good prediction effect. However, ELMs ignore the inherent distribution of the original samples, and they are prone to overfitting, which fails at achieving good generalization performance. In this paper, based on expectile penalty and correntropy, an asymmetric C-loss function (called AC-loss) is proposed, which is non-convex, bounded, and relatively insensitive to noise. Further, a novel extreme learning machine called L$_1$ norm robust regularized extreme learning machine with asymmetric C-loss (L$_1$-ACELM) is presented to handle the overfitting problem. The proposed algorithm benefits from L$_1$ norm and replaces the square loss function with the AC-loss function. The L$_1$-ACELM can generate a more compact network with fewer hidden nodes and reduce the impact of noise. To evaluate the effectiveness of the proposed algorithm on noisy datasets, different levels of noise are added in numerical experiments. The results for different types of artificial and benchmark datasets demonstrate that L$_1$-ACELM achieves better generalization performance compared to other state-of-the-art algorithms, especially when noise exists in the datasets.

**Keywords:** extreme learning machine; asymmetric least square loss; expectile; correntropy; robustness

## 1. Introduction

The single hidden-layer feedforward neural network (SLFN) is one of the most important learning algorithms in data mining and machine learning fields. SLFN has only one hidden layer that connects the input and output layers. Generally, gradient-based algorithms are used to train SLFNs similar to back-propagation algorithms [1], which often leads to slow convergence, overfitting, and local minima. To overcome these problems, Huang et al. [2,3] proposed a widely used method based on the structure of SLFN called extreme learning machine (ELM). Compared to the traditional single hidden layer feedforward neural network, the input weights and thresholds of the hidden layer nodes in ELM are randomly generated, and there is no need for repeated adjustment via iterations. ELM identifies the output weight vector with the smallest norm by calculating the Moore-Penrose inverse. Therefore, the training speed of ELM is much higher than that of SLFN. Moreover, ELM also requires minimal training error and norm of the weights, which facilitates good generalization performance. Since ELM has a higher learning speed and better generalization performance, it has been successfully applied in many fields [4–6]. However, ELM still has several shortcomings. For example, ELM is based on empirical risk minimization (ERM) [7] which often leads to overfitting.

To address this issue, many scholars have proposed various algorithms based on ELM to improve the generalization performance. In [8], Deng et al. introduced the weight factor $\gamma$ into ELM for the first time and proposed the regularized extreme learning machine (RELM). By adjusting the weight factor $\gamma$, the proportion of empirical risk and structural risk

in the actual prediction risk can be optimal, thereby avoiding model overfitting. However, RELM uses the $L_2$ norm which is sensitive to outliers. To reduce the influence of outliers, Rong et al. proposed the pruned extreme learning machine (P-ELM) [9], which can remove irrelevant hidden nodes. P-ELM is only used for classification problems. To further address the regression problem, the optimally pruned extreme learning machine (OP-ELM) [10] was proposed. In OP-ELM, The $L_1$ norm is used to remove irrelevant output nodes and select the corresponding hidden nodes, and then the weight of the corresponding hidden nodes is calculated using the least squares method. Given that the $L_1$ norm is robust to outliers, it is used in various algorithms to improve the generalization performance [11,12]. Balasundaram et al. [13] proposed the $L_1$ norm extreme learning machine, which produces sparse models such that decision functions can be determined using fewer hidden layer nodes. Generally speaking, RELM is composed of empirical risk and structural risk. Structural risk can effectively avoid overfitting, and structural risk is determined by loss function. Traditional RELMs use the squared loss function, which is symmetric and unbounded. The symmetry makes the model unable to take into account the distribution characteristics within the training samples, while unboundedness will cause the model to be sensitive to noise and outliers. In real life, the distribution of data is unbalanced, and noise is generally mixed in the process of data collection. Therefore, it is particularly important to choose an appropriate loss function to construct the model.

Quantiles can reflect completely the distribution of random variables without missing any information Quantile regression can more accurately describe the distribution characteristics of random variables for comprehensive analysis. Therefore, quantile regression is more robust and has been successfully applied to statistical prediction [14,15]. Quantile loss can be thought of as a pinball penalty. Expectile loss is an asymmetric least squares loss, which is the square of the quantile loss function. It is often used in regression problems with imbalanced data [16]. However, the unboundedness of the expectile loss leads to a lack of robustness.

From [17], the bounded loss function is less sensitive to noise and outliers than the unbounded loss function, whereas convex functions are usually unbounded. To further improve the robustness of ELM, researchers have proposed various non-convex loss functions to replace the convex loss functions [18–20]. Examples of common convex loss functions include square loss, hinge loss, and Huber loss, which allow for the determination of global optimal solutions and are easy to solve. However, the unboundedness of the convex loss function implies that it is not suited for handling outliers. Compared to convex loss functions, non-convex loss functions are more robust to outliers. Recently, Singh et al. [21] proposed a correntropy-based loss function called C-loss. Based on information theory and the kernel method, correntropy [22,23] is considered to be a generalized local similarity measure between two random variables. As a non-convex, bounded loss function, the C-loss function has been widely used in machine learning to improve robustness. In 2019, Zhao et al. [24] applied the C-loss function to ELM for the first time. They proposed the C-loss based ELM (CELM), and also experimentally demonstrated that the generalization performance was better compared to that of other algorithms.

In real life, the distribution of datasets tends to be asymmetric, and the training samples are easily contaminated by noise. In order to better consider the distribution characteristics inside the data and improve the generalization ability of the algorithm, a non-convex robust loss function is proposed, called asymmetric C-loss (AC-loss). A robust extreme learning machine based on the asymmetric C-loss and $L_1$-norm (called $L_1$-ACELM) is then developed. The main contributions of this report are as follows:

(1) Based on the expectile penalty and correntropy loss function, a new loss function (AC-loss) is developed. AC-loss retains some important properties of C-loss such as non-convexity and boundedness. AC-loss is asymmetric, and it can handle unbalanced noise.

(2) A novel approach called the $L_1$-norm robust regularized extreme learning machine with asymmetric C-loss ($L_1$-ACELM) is proposed by applying the proposed AC-loss

function and the $L_1$-norm in the objective function of ELM to enhance robustness to outliers.

(3)  The non-convexity of the AC-loss function makes it difficult for $L_1$-ACELM to be solved. The half-quadratic optimization algorithm [25–27] is used to address these problems. Moreover, the convergence of the proposed algorithms is analyzed.

The remainder of this paper is structured as follows. Section 2 briefly reviews ELM, RELM, C-loss function, and the half-quadratic optimization algorithm. In Section 3, we propose the asymmetric C-loss function and the $L_1$-ACELM model. Next, the half-quadratic optimization algorithm is used to solve $L_1$-ACELM. In addition, we analyze the convergence of the algorithm. The experimental results for the artificial and benchmark datasets are presented in Section 4. Section 5 summarizes the main conclusions and further study.

## 2. Related Work

### 2.1. Extreme Learning Machine (ELM)

ELM is a new single hidden layer feedforward neural network that is first proposed by Huang et al. [2]. Unlike traditional SLFN, the input weights and thresholds of the hidden layer in ELM are randomly generated and the output weights can be determined using the least square method. Hence, it is much faster than traditional SLFN. In addition, ELM has good generalization ability.

Given $N$ arbitrary distinct samples $\{X, Y\} = \{x_i, y_i\}_{i=1}^N, x_i = [x_{i1}, x_{i2}, \ldots, x_{im}]^T \in \mathbb{R}^m$ and $y_i = [y_{i1}, y_{i2}, \ldots, y_{in}]^T \in \mathbb{R}^n$ are the input samples and the corresponding output vectors, respectively. The output of a standard SLFN with $L$ hidden nodes can be expressed as follows:

$$f(x_i) = \sum_{j=1}^L \beta_j h(\alpha_j, b_j, x_i), i = 1, \ldots, N \tag{1}$$

where $\alpha_j = [\alpha_{j1}, \alpha_{j2}, \ldots, \alpha_{jm}]^T \in \mathbb{R}^m$ is the input weight vector that connects the input node to the $j$-th hidden layer node and $b_j \in R$ is the bias of the $j$-th hidden node. $\beta_j = [\beta_{j1}, \beta_{j2}, \ldots, \beta_{jn}]^T \in \mathbb{R}^n$ is the output weight vector that connects the $j$-th hidden layer node to the output node, and $h(\alpha_j, b_j, x_i)$ is the output of the $j$-th hidden layer node with respect to the input $x_i$. $f(\cdot)$ denotes the actual output vector of SLFN.

For ELM, the input weight vector and the bias that connects the input node to the hidden layer node are randomly assigned instead of being updated. Therefore, it can be converted to a linear model:

$$F = H\beta \tag{2}$$

where

$$H = \begin{bmatrix} h(x_1) \\ \vdots \\ h(x_N) \end{bmatrix} = \begin{bmatrix} h(\alpha_1, b_1, x_1) & \ldots & h(\alpha_L, b_L, x_1) \\ \vdots & \ddots & \vdots \\ h(\alpha_1, b_1, x_N) & \ldots & h(\alpha_L, b_L, x_N) \end{bmatrix}_{N \times L}, \beta = \begin{bmatrix} \beta_1^T \\ \vdots \\ \beta_L^T \end{bmatrix}_{L \times n} \text{ and } F = \begin{bmatrix} f(x_1)^T \\ \vdots \\ f(x_N)^T \end{bmatrix}_{N \times n}$$

Here, $H$ is the output matrix of the hidden layer. Thus, the output weight vector that connects the hidden layer node to the output node can be determined by solving the following equation:

$$\min_\beta \|H\beta - Y\|_2 \tag{3}$$

ELM requires the approximation of the training samples with zero error. Therefore, Equation (3) can be written as:

$$H\beta = Y \tag{4}$$

The output weight $\beta$ is the least squares solution of Equation (4), which can be obtained as follows:

$$\beta = H^+ Y \tag{5}$$

where $H^+$ is the Moore-Penrose generalized inverse of the matrix $H$.

To avoid overfitting of the model, regularized ELM is proposed, which facilitates better generalization performance by minimizing the sum of the training error and the norm of the output weights [28]. RELM can be expressed as follows:

$$\min_{\boldsymbol{\beta}} \|H\boldsymbol{\beta} - \boldsymbol{Y}\|_2^2 + \frac{\gamma}{2}\|\boldsymbol{\beta}\|_2^2 \tag{6}$$

The optimal solution to RELM is computed as follows:

$$\boldsymbol{\beta} = \begin{cases} \left(H^T H + \gamma I\right)^{-1} H^T \boldsymbol{Y} & if\ N \geq L \\ H^T \left(H H^T + \gamma I\right)^{-1} \boldsymbol{Y} & if\ N < L \end{cases} \tag{7}$$

where $I$ is an identity matrix.

### 2.2. Correntropy-Induced Loss (C-Loss)

Correntropy is a generalized similarity measure between two random variables in a small neighborhood defined by the kernel width $\sigma$. For a regression problem, the choice of the loss function could ensure that the similarity between the actual output and the target value is maximized, which is equivalent to the maximization of correntropy. Thus, the C-loss function [21] is proposed by Singh et al., which is defined as:

$$L_C(y_i, f(\boldsymbol{x}_i)) = 1 - \exp\left\{-\frac{(y_i - f(\boldsymbol{x}_i))^2}{2\sigma^2}\right\} \tag{8}$$

As a bounded non-convex loss function, the C-loss loss function is more robust to outliers than the traditional squared loss function.

### 2.3. Half-Quadratic Optimization

The half-quadratic optimization algorithm based on the conjugate function theory [29] is usually used for convex optimization and non-convex optimization problems. This method transforms the original non-convex objective function into a half-quadratic objective function by introducing auxiliary variables. As such, the objective function cannot be solved directly, and a two-step alternating minimization method is required. The specific operations are as follows: given the original variables, the auxiliary variables are optimized. The variables are then optimized, and the original variables are determined.

The minimization problem is as follows:

$$\min_{\boldsymbol{v}} \phi_{\boldsymbol{v}}(\boldsymbol{v}) + F(\boldsymbol{v}) \tag{9}$$

where $\boldsymbol{v} = [v_1, v_2, \ldots, v_N]^T \in \mathbb{R}^N$, $\phi(\cdot)$ is a potential loss function with $\phi(\boldsymbol{v}) = \sum\limits_{i=1}^{N} \phi(v_i)$ and $F(\cdot)$ is a convex penalty function.

Considering the half-quadratic optimization algorithm, we introduce an auxiliary variable $\boldsymbol{p} = [p_1, p_2, \ldots, p_N]^T \in \mathbb{R}^N$ into $\phi(\cdot)$, which can then be expressed as:

$$\phi(v_i) = \min_{p_i}\{Q(v_i, p_i) + \varphi(p_i)\} \tag{10}$$

where $Q(v_i, p_i)$ is a half-quadratic function, which can be represented in the additive form $Q_A(v_i, p_i) = \frac{1}{2}\left(\sqrt{c}v_i - p_i/\sqrt{c}\right)^2$ or the multiplicative form $Q_M(v_i, p_i) = \frac{1}{2}p_i v_i^2$.

Substituting Equation (10) into Equation (9), we obtain the following optimization problem:

$$\min_{\boldsymbol{v}} \phi_{\boldsymbol{v}}(\boldsymbol{v}) + F(\boldsymbol{v}) = \min_{\boldsymbol{v},\boldsymbol{p}}\{Q(\boldsymbol{v},\boldsymbol{p}) + \varphi(\boldsymbol{p}) + F(\boldsymbol{v})\} \tag{11}$$

where $p_i$ is determined using a function $g(\cdot)$, which is the conjugate function of $\phi(\cdot)$. Alternatively, Equation (11) can then be optimized as follows:

$$p^{t+1} = g(v) \tag{12}$$

$$v^{t+1} = \underset{v}{\arg\min}\left\{ Q\left(v, p^{t+1}\right) + F(v)\right\} \tag{13}$$

where $t$ represents the $t$-th iteration.

## 3. Main Contributions

### 3.1. Asymmetric C-Loss Function (AC-Loss)

As a measure of risk, the expectile is an extension of the quantile, which represents the distributional information of a random variable. The expectile loss is essentially a squared pinball loss, which can also be considered as an asymmetric squared loss. The asymmetric least square loss function can be expressed as:

$$L_\tau(y_i, f(x_i)) = \begin{cases} \tau(y_i - f(x_i))^2 & if\ y_i - f(x_i) \geq 0 \\ (1-\tau)(y_i - f(x_i))^2 & if\ y_i - f(x_i) < 0 \end{cases} \tag{14}$$

However, given that the asymmetric least square loss is an unbounded loss function, it is more sensitive to outliers. Therefore, we construct an asymmetric C-loss (AC-loss) function, based on the C-loss function and the expectile loss function, which is a non-convex, asymmetric, and bounded function for dealing with outliers and noise. The AC-loss function is defined as follows:

$$L_C^{als}(y_i, f(x_i)) = \begin{cases} 1 - \exp\left\{\dfrac{-\tau(y_i - f(x_i))^2}{2\sigma^2}\right\} & if\ y_i - f(x_i) \geq 0 \\ 1 - \exp\left\{\dfrac{-(1-\tau)(y_i - f(x_i))^2}{2\sigma^2}\right\} & if\ y_i - f(x_i) < 0 \end{cases} \tag{15}$$

The plot of the AC-loss function is shown in Figure 1.

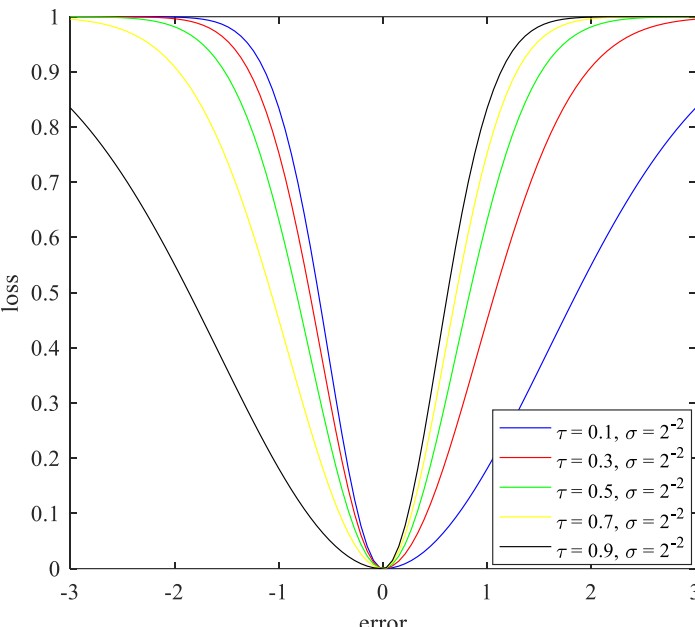

**Figure 1.** Asymmetric C-loss function.

### 3.2. L₁-ACELM

To improve the generalization performance of RELM, the proposed loss function is introduced to replace the squared loss function. To further enhance robustness to outliers, the L$_2$ norm of structural risk in RELM is replaced with the L$_1$ norm. Therefore, we propose a new robust ELM (called L$_1$-ACELM):

$$\min_{\boldsymbol{\beta}} J(\boldsymbol{\beta}) = \sum_{i=1}^{N} L_C^{als}(y_i - \boldsymbol{h}(\boldsymbol{x}_i)\boldsymbol{\beta}) + \gamma\|\boldsymbol{\beta}\|_1 \tag{16}$$

where $\gamma > 0$ is a regularized parameter.

Since AC-loss is a non-convex loss function, it is difficult to directly optimize the objective function. The half-quadratic optimization algorithm is usually applied to optimize non-convex problems. Therefore, we chose the half-quadratic optimization algorithm to find the optimal solution of the objective function.

### 3.3. Solving Method

For the function $f(u) = \exp(u)$, there exists a convex function $g(v)$, which is expressed as follows:

$$g(v) = -v\log(-v) + v \tag{17}$$

where $v < 0$, and the conjugate function $g^*(u)$ of the function $g(v)$ is defined as:

$$g^*(u) = \sup_{v}\{uv + v\log(-v) - v\} \tag{18}$$

where

$$v = -\exp(-u) < 0 \tag{19}$$

By substituting Equation (19) into Equation (18), we have

$$g^*(u) = \exp(-u) \tag{20}$$

Now, let $u = \begin{cases} \frac{\tau e_i^2}{2\sigma^2} & if\ e_i \geq 0 \\ \frac{(1-\tau)e_i^2}{2\sigma^2} & if\ e_i < 0 \end{cases}$ and $e_i = y_i - \boldsymbol{h}(\boldsymbol{x}_i)\boldsymbol{\beta}$, then Equation (18) can be expressed as:

$$g^*(u) = \begin{cases} \sup_{v}\left\{\frac{\tau e_i^2}{2\sigma^2}v + v\log(-v) - v\right\} \\ \sup_{v}\left\{\frac{(1-\tau)e_i^2}{2\sigma^2}v + v\log(-v) - v\right\} \end{cases} = \begin{cases} \exp\left(-\frac{\tau e_i^2}{2\sigma^2}\right) & if\ e_i \geq 0 \\ \exp\left(-\frac{(1-\tau)e_i^2}{2\sigma^2}\right) & if\ e_i < 0 \end{cases} \tag{21}$$

where

$$v_i = \begin{cases} -\exp\left(-\frac{\tau e_i^2}{2\sigma^2}\right) & if\ e_i \geq 0 \\ -\exp\left(-\frac{(1-\tau)e_i^2}{2\sigma^2}\right) & if\ e_i < 0 \end{cases} \tag{22}$$

By combining Equations (21) and (16), we have

$$\min_{\boldsymbol{\beta},v} J(\boldsymbol{\beta},v) = \begin{cases} \sum_{i=1}^{N}\left(1 - \sup_{v_i}\left\{\exp\left(-\frac{\tau e_i^2}{2\sigma^2}\right)v_i + g(v_i)\right\}\right) + \gamma\|\boldsymbol{\beta}\|_1 & if\ e_i \geq 0 \\ \sum_{i=1}^{N}\left(1 - \sup_{v_i}\left\{\exp\left(-\frac{(1-\tau)e_i^2}{2\sigma^2}\right)v_i + g(v_i)\right\}\right) + \gamma\|\boldsymbol{\beta}\|_1 & if\ e_i < 0 \end{cases} \tag{23}$$
$$s.t.\ \boldsymbol{\beta}h(\boldsymbol{x}_i) = y_i - e_i, i = 1,2,\ldots,N$$

where $v = [v_1, v_2, \ldots, v_N]^T$. Equation (23) can be simplified as:

$$
\min_{\beta, v} J'(\beta, v) = \begin{cases} \sup_v \left\{ \sum_{i=1}^N \left( -\frac{\tau e_i^2}{2\sigma^2} v_i - v_i \log(-v_i) + v_i \right) \right\} + \gamma \|\beta\|_1 & if\ e_i \geq 0 \\ \sup_v \left\{ \sum_{i=1}^N \left( -\frac{(1-\tau)e_i^2}{2\sigma^2} v_i - v_i \log(-v_i) + v_i \right) \right\} + \gamma \|\beta\|_1 & if\ e_i < 0 \\ s.t.\ h(x_i)\beta = y_i - e_i,\ i = 1, 2, \ldots, N \end{cases} \tag{24}
$$

The optimal solution $\beta$ can be obtained by solving Equation (24) using the alternating optimization method.

Firstly, given the original variables $\beta^t$, we can obtain the optimal solution for the auxiliary variables $v^{t+1}$. When $\beta^t$ is given, the minimization problem is given as follows:

$$
\min_v J(v) = \begin{cases} \sum_{i=1}^N \left( -\frac{\tau(y_i - f(x_i))^2}{2\sigma^2} v_i - v_i \log(-v_i) + v_i \right) & if\ e_i \geq 0 \\ \sum_{i=1}^N \left( -\frac{(1-\tau)(y_i - f(x_i))^2}{2\sigma^2} v_i - v_i \log(-v_i) + v_i \right) & if\ e_i < 0 \end{cases} \tag{25}
$$

According to the half-quadratic optimization algorithm, the auxiliary variables $v^{t+1}$ can be obtained by solving Equation (24). Thus, we have:

$$
v_i^{t+1} = \begin{cases} -\exp\left( -\frac{\tau\left(y_i - f^t(x_i)\right)^2}{2\sigma^2} \right) & if\ e_i \geq 0 \\ -\exp\left( -\frac{(1-\tau)\left(y_i - f^t(x_i)\right)^2}{2\sigma^2} \right) & if\ e_i < 0 \end{cases}, i = 1, 2, \ldots, N \tag{26}
$$

Secondly, the auxiliary variables $v^{t+1}$ are fixed and the optimal solution of the original variable $\beta^{t+1}$ can be obtained by solving the following minimization problem:

$$
\min_{\beta^{t+1}} J(\beta^{t+1}) = \begin{cases} \sum_{i=1}^N \left( -\frac{\tau v_i}{2\sigma^2} e_i^2 \right) + \gamma \|\beta^{t+1}\|_1 & if\ e_i \geq 0 \\ \sum_{i=1}^N \left( -\frac{(1-\tau)v_i}{2\sigma^2} e_i^2 \right) + \gamma \|\beta^{t+1}\|_1 & if\ e_i < 0 \\ s.t.\ \beta^{t+1}h(x_i) = y_i - e_i, i = 1, 2, \ldots, N \end{cases} \tag{27}
$$

Equation (27) is equivalent to

$$
\min_{\beta^{t+1}} J(\beta^{t+1}) = \begin{cases} \sum_{i=1}^N \left( -\frac{\tau v_i^{t+1}}{2\sigma^2} \left( y_i - h(x_i)\beta^{t+1} \right)^2 \right) + \gamma \|\beta^{t+1}\|_1 & if\ y_i \geq h(x_i)\beta^{t+1} \\ \sum_{i=1}^N \left( -\frac{(1-\tau)v_i^{t+1}}{2\sigma^2} \left( y_i - h(x_i)\beta^{t+1} \right)^2 \right) + \gamma \|\beta^{t+1}\|_1 & if\ y_i < h(x_i)\beta^{t+1} \end{cases} \tag{28}
$$

Since the $L_1$ norm exists in the objective function, the proximal gradient descent (PGD) algorithm is applied to solve the optimization problem Equation (28). The objective function $J(\beta^{t+1})$ can be written as

$$
J\left(\beta^{t+1}\right) = S\left(\beta^{t+1}\right) + \gamma \left\|\beta^{t+1}\right\|_1, \tag{29}
$$

where

$$
S\left(\beta^{t+1}\right) = \begin{cases} \sum_{i=1}^N \left( -\frac{\tau v_i^{t+1}}{2\sigma^2} \left( y_i - h(x_i)\beta^{t+1} \right)^2 \right) & if\ y_i \geq h(x_i)\beta^{t+1} \\ \sum_{i=1}^N \left( -\frac{(1-\tau)v_i^{t+1}}{2\sigma^2} \left( y_i - h(x_i)\beta^{t+1} \right)^2 \right) & if\ y_i < h(x_i)\beta^{t+1} \end{cases} \tag{30}
$$

$S(\boldsymbol{\beta}^{t+1})$ is differentiable and its derivative is as follows:

$$\nabla S(\boldsymbol{\beta}^{t+1}) = \begin{cases} \sum\limits_{i=1}^{N}\left(\frac{\tau v_i^{t+1}}{\sigma^2}\boldsymbol{h}^T(\boldsymbol{x}_i)(y_i - \boldsymbol{h}(\boldsymbol{x}_i)\boldsymbol{\beta}^{t+1})\right) & if \ y_i \geq \boldsymbol{h}(\boldsymbol{x}_i)\boldsymbol{\beta}^{t+1} \\ \sum\limits_{i=1}^{N}\left(\frac{(1-\tau)v_i^{t+1}}{\sigma^2}\boldsymbol{h}^T(\boldsymbol{x}_i)(y_i - \boldsymbol{h}(\boldsymbol{x}_i)\boldsymbol{\beta}^{t+1})\right) & if \ y_i < \boldsymbol{h}(\boldsymbol{x}_i)\boldsymbol{\beta}^{t+1} \end{cases} \quad (31)$$

Since $\nabla S(\boldsymbol{\beta}^{t+1})$ satisfies the L-Lipschitz continuity condition, there is a constant $\eta > 0$ such that

$$\left\|\nabla S(\boldsymbol{\beta}) - \nabla S(\boldsymbol{\beta}^{t+1})\right\|_2^2 \leq \eta\left\|\boldsymbol{\beta} - \boldsymbol{\beta}^{t+1}\right\|_2^2, \forall(\boldsymbol{\beta}, \boldsymbol{\beta}^{t+1}) \quad (32)$$

The second-order Taylor expansion of the function $S(\boldsymbol{\beta}^{t+1})$ can be expressed as

$$\begin{aligned} S(\boldsymbol{\beta}; \boldsymbol{\beta}^{t+1}) &\approx S(\boldsymbol{\beta}^{k+1}) + \nabla S(\boldsymbol{\beta}^{k+1})(\boldsymbol{\beta} - \boldsymbol{\beta}^{k+1}) + \frac{\eta}{2}\left\|\boldsymbol{\beta} - \boldsymbol{\beta}^{k+1}\right\| \\ &= \frac{\eta}{2}\left\|\boldsymbol{\beta} - \left(\boldsymbol{\beta}^{k+1} - \frac{1}{\eta}\nabla S(\boldsymbol{\beta}^{k+1})\right)\right\|_2^2 + \delta(\boldsymbol{\beta}^{k+1}) \end{aligned} \quad (33)$$

where $\delta(\boldsymbol{\beta}^{t+1})$ is a constant that is independent of $\boldsymbol{\beta}^{t+1}$.

Introducing $\left\|\boldsymbol{\beta}^{t+1}\right\|_1$ into the objective function, the iterative equation of the proximal gradient descent can be expressed as

$$\boldsymbol{\beta}^{t+1} = \underset{\boldsymbol{\beta}^{t+1}}{\operatorname{argmin}}\frac{\eta}{2}\left\|\boldsymbol{\beta} - \left(\boldsymbol{\beta}^{t+1} - \frac{1}{\eta}\nabla S(\boldsymbol{\beta}^{t+1})\right)\right\|_2^2 + \gamma\left\|\boldsymbol{\beta}^{t+1}\right\|_1 \quad (34)$$

Let $\boldsymbol{z} = \boldsymbol{\beta}^{t+1} - \frac{1}{\eta}\nabla S(\boldsymbol{\beta}^{t+1})$. Then, the closed-form solution of Equation (34) can be written as:

$$\boldsymbol{\beta}_i^{t+1} = \begin{cases} z_i - \gamma/\eta & \gamma/\eta < z_i \\ 0 & |z_i| \leq \gamma/\eta \\ z_i + \gamma/\eta & z_i < -\gamma/\eta \end{cases}, i = 1, 2, \ldots, N \quad (35)$$

where $\boldsymbol{\beta}_i^{t+1}$ and $z_i$ represent the $i$-th component of $\boldsymbol{\beta}^{t+1}$ and $\boldsymbol{z}$, respectively. We develop a half-quadratic optimization to solve the proposed model, and the pseudo code is presented in Algorithm 1.

---

**Algorithm 1. Half-quadratic optimization for L$_1$-ACELM**

---

Input: The training dataset $T = \{(x_i, y_i)\}_{i=1}^{N}$, the number of hidden layer nodes $L$, the activation function $h(\boldsymbol{x})$, the regularization parameter $\gamma$, the maximum number of iterations $t_{\max}$, window width $\sigma$, a small number $\rho$ and the parameter $\tau$.
Output: the output weight vector $\boldsymbol{\beta}$.
Step 1. Randomly generate input weight $\boldsymbol{\alpha}_i$ and hidden layer bias $b_i$ with $L$ hidden nodes.
Step 2. Calculate hidden output matrix $H(\boldsymbol{x})$.
Step 3. Compute $\boldsymbol{\beta}$ by Equation (7).
Step 4. Let $\boldsymbol{\beta}^0 = \boldsymbol{\beta}$ and $\boldsymbol{\beta}^1 = \boldsymbol{\beta}$, set $t = 1$.
Step 5. While $|J(\boldsymbol{\beta}^t) - J(\boldsymbol{\beta}^{t-1})| < \rho$ or $t < t_{\max}$ do
calculate $v_i^{t+1}$ by Equation (26).
update $\boldsymbol{\beta}^{t+1}$ using Equation (35).
compute $J(\boldsymbol{\beta}^{t+1})$ by Equation (29).
update $t := t + 1$.
End while
Step 6: Output result given by $\boldsymbol{\beta} = \boldsymbol{\beta}^{t-1}$.

---

*3.4. Convergence Analysis*

**Proposition 1.** *The sequence* $\left\{ J\left( \boldsymbol{\beta}^t, \boldsymbol{v}^t \right), t = 1, 2, \ldots, t \right\}$ *generated by Algorithm 1 is convergent.*

**Proof.** Let $\boldsymbol{\beta}^t$ and $\boldsymbol{v}^t$ be the optimal solution to the objective function (23) after $t$ iterations. In the half-quadratic optimization problem, the conjugate function $g^*(\cdot)$ satisfies $\{Q(\boldsymbol{\beta}_i, g^*(\boldsymbol{\beta}_i)) + \varphi(\boldsymbol{\beta}_i)\} \leq \{Q(\boldsymbol{\beta}_i, g^*(v_i)) + \varphi(v_i)\}$. When $\boldsymbol{\beta}^t$ is fixed, we can obtain the optimal solution $\boldsymbol{v}^{t+1}$ of $\boldsymbol{v}$ at the $(t + 1)$-th iteration from Equation (26), then we have:

$$J\left( \boldsymbol{\beta}^t, \boldsymbol{v}^{t+1} \right) \leq J\left( \boldsymbol{\beta}^t, \boldsymbol{v}^t \right) \tag{36}$$

Next, when $\boldsymbol{v}^{t+1}$ is fixed, we can optimize (28) to obtain the solution $\boldsymbol{\beta}^{t+1}$ of $\boldsymbol{\beta}$ at the $(t + 1)$-th iteration. Then we have:

$$J\left( \boldsymbol{\beta}^{t+1}, \boldsymbol{v}^{t+1} \right) \leq J\left( \boldsymbol{\beta}^t, \boldsymbol{v}^{t+1} \right) \tag{37}$$

Combining Inequation (36) with Inequality (37), we have:

$$J\left( \boldsymbol{\beta}^{t+1}, \boldsymbol{v}^{t+1} \right) \leq J\left( \boldsymbol{\beta}^t, \boldsymbol{v}^{t+1} \right) \leq J\left( \boldsymbol{\beta}^t, \boldsymbol{v}^t \right) \tag{38}$$

Hence, the optimization problem $J(\boldsymbol{\beta}, \boldsymbol{v})$ is bounded, and the sequence $\left\{ J\left( \boldsymbol{\beta}^t, \boldsymbol{v}^t \right), t = 1, 2, \ldots, t \right\}$ is convergent. □

## 4. Experiments

*4.1. Experimental Setup*

To evaluate the performance of the proposed $L_1$-ACELM algorithm, we performed numerical simulations using two artificial datasets and ten standard benchmark datasets. To show the effectiveness of the $L_1$-ACELM algorithm compared to traditional algorithms including extreme learning machine (ELM), regularized ELM (RELM), and C-loss based ELM (CELM), several experiments were performed. All experiments were implemented in Matlab2016a on a PC with an i5-7200U Intel(R) Core (TM) processor (2.70 GHz) 4 GB RAM.

To evaluate the prediction performance of the $L_1$-ACELM algorithm, the regression evaluation metrics are defined as follows:

(1) The root mean square error (*RMSE*)

$$RMSE = \sqrt{\frac{1}{N} \sum_{i=1}^{N} (y_i - \hat{y}_i)^2} \tag{39}$$

(2) Mean absolute error (*MAE*)

$$MAE = \frac{1}{N} \sum_{i=1}^{N} |y_i - \hat{y}_i| \tag{40}$$

(3) The ratio of the sum squared error (*SSE*) to the sum squared deviation of the sample *SST* (*SSE/SST*) is given as:

$$SSE/SST = \frac{\sum_{i=1}^{N} (\hat{y}_i - y_i)^2}{\sum_{i=1}^{N} (y_i - \overline{y}_i)^2} \tag{41}$$

(4) The ratio between the interpretable sum deviation *SSR* and *SST* (*SSR/SST*) is given by:

$$SSR/SST \; = \; \frac{\sum\limits_{i=1}^{N}\left(\hat{y}_i \; - \; \overline{y}_i\right)^2}{\sum\limits_{i=1}^{N}\left(y_i \; - \; \overline{y}_i\right)^2} \tag{42}$$

where $N$ is the number of samples. $y_i$ and $\hat{y}_i$ denote the target values and the corresponding predicted values, respectively. $\overline{y}_i$ can be calculated from $\overline{y}_i \; = \; \frac{1}{N}\sum\limits_{i=1}^{N} y_i$, which represents the average value of $y_1, y_2, \ldots, y_N$. The sigmoid function is chosen as the activation function for ELM, RELM, CELM, and L$_1$-ACELM, and can be expressed as:

$$h(x) \; = \; \frac{1}{1 + \exp\left(-a_i^T x + b_i\right)} \tag{43}$$

Since the original algorithms and the proposed algorithm involve many parameters, to ensure the best performance, ten-fold cross-validation is used to determine the optimal parameters. In ELM and RELM, the number of hidden layer nodes $L \; = \; 30$ is fixed. For RELM, CELM, and L$_1$-ACELM, the optimal value of the regularization parameter $\gamma$ is selected from the set $\{2^{-50}, 2^{-49}, \ldots, 2^{49}, 2^{50}\}$. For CELM and L$_1$-ACELM, the window width $\sigma$ is selected from the range $\{2^{-2}, 2^{-1}, 2^0, 2^1, 2^2\}$. For L$_1$-ACELM, the parameter $\tau$ is obtained from the set $\{0.1, 0.2, \ldots, 0.9\}$.

### 4.2. Performance on Artificial Datasets

To verify the robustness of the proposed L$_1$-ACELM, two artificial datasets were generated using six different types of noise, both of which consisted of 2000 data points. Table 1 shows the specific forms of two artificial datasets and different types of noise. $\lambda_i \sim N\left(0, s^2\right)$ indicates that $\lambda_i$ has a normal distribution with a mean of zero and variance of $s^2$, $\lambda_i \sim U(a, b)$ means that $\lambda_i$ has a uniform distribution in the interval $[a, b]$, $\lambda_i \sim T(c)$ indicates that $\lambda_i$ has a t-distribution with $c$ degrees of freedom.

**Table 1.** Artificial datasets with different types of noise.

| Artificial Dataset | Function Definition | Types of Noise |
|---|---|---|
| Sinc function | $y_i \; = \; \mathrm{sin}\,c(2x_i) \; = \; \frac{\sin(2x_i)}{2x_i} + \lambda_i$ | Type A: $x \in [-3, 3], \lambda_i \sim N(0, 0.15^2)$ |
| | | Type B: $x \in [-3, 3], \lambda_i \sim N(0, 0.5^2)$ |
| | | Type C: $x \in [-3, 3], \lambda_i \sim U(-0.15, 0.15)$ |
| Self-defining function | $y_i \; = \; e^{x_i^2 \, \mathrm{sin}\,c(0.3\pi x_i)} + \lambda_i$ | Type D: $x \in [-3, 3], \lambda_i \sim U(0.5, 0.5)$ |
| | | Type E: $x \in [-3, 3], \lambda_i \sim T(5)$ |
| | | Type F: $x \in [-3, 3], \lambda_i \sim T(10)$ |

Figure 2 shows different types of noise graphs, the graphs of the sinc function, and the graphs of the sinc function with different noises.

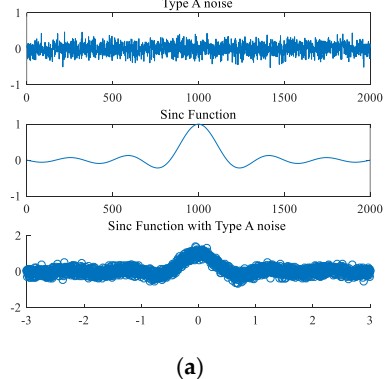

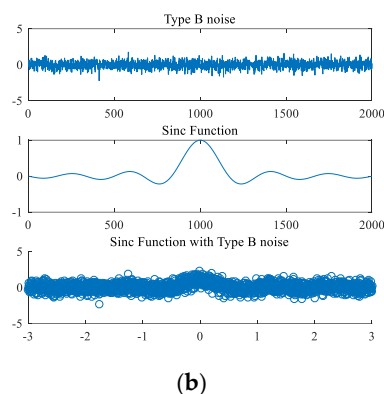

(a)　　　　　　　　　　　　　　　　　　　(b)

**Figure 2.** *Cont.*

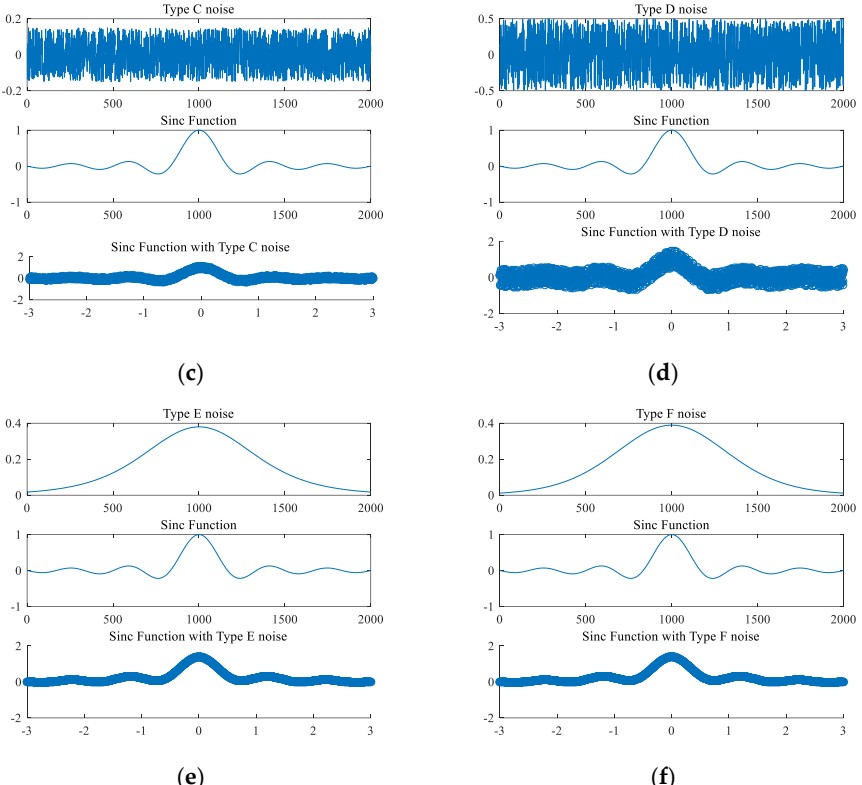

**Figure 2.** Graphs of the sinc function with different noises.

Figure 3 shows different types of noise graphs, the graphs of the self-defining function, and the graphs of the self-defining function with different noises.

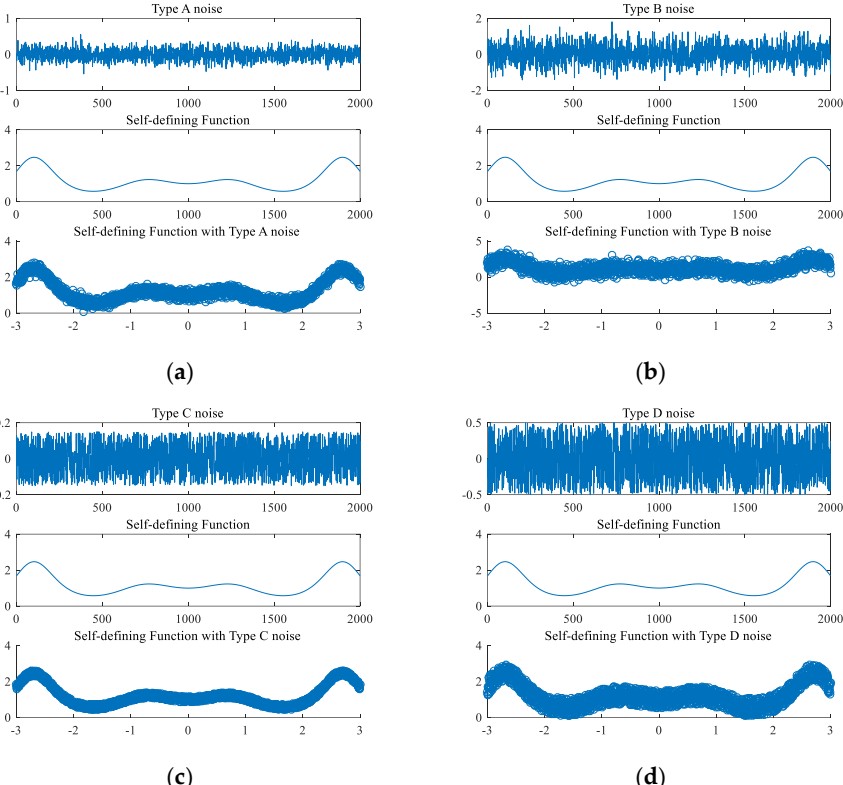

**Figure 3.** *Cont.*

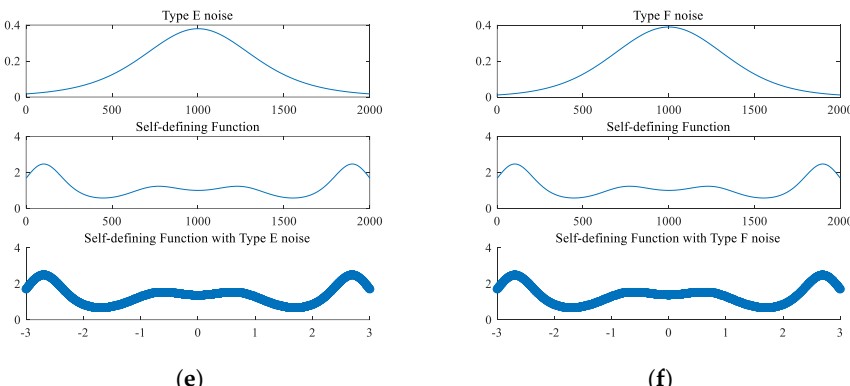

**Figure 3.** Graphs of the self-defining function with different noises.

In our experiments, we randomly selected 1600 samples as the training dataset and the remaining 400 samples as the testing dataset. To evaluate the effectiveness of the proposed algorithm, we compared its performance to that of ELM, RELM, and CELM. Table 2 shows the optimal RMSE, MAE, SSE/SST, and SSR/SST of the four algorithms that were obtained based on the optimal parameters selected using the ten-fold cross-validation method. Table 2 also lists the optimal parameters for each algorithm. The regression fitting results of ELM, RELM, CELM, and $L_1$-ACELM on two artificial datasets with noise are shown in Figures 4 and 5.

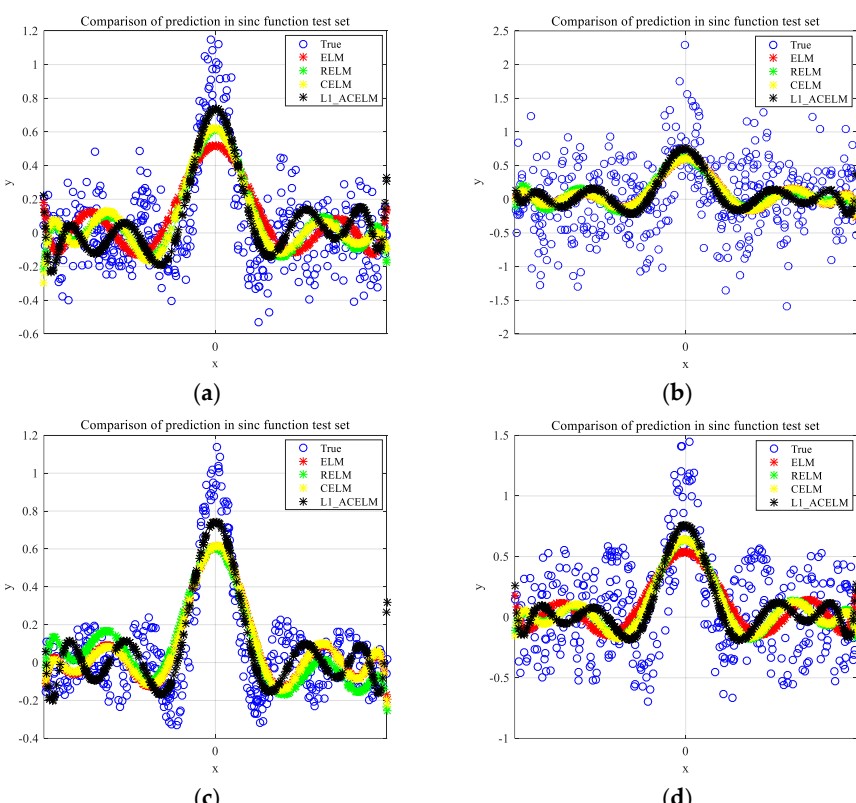

**Figure 4.** *Cont.*

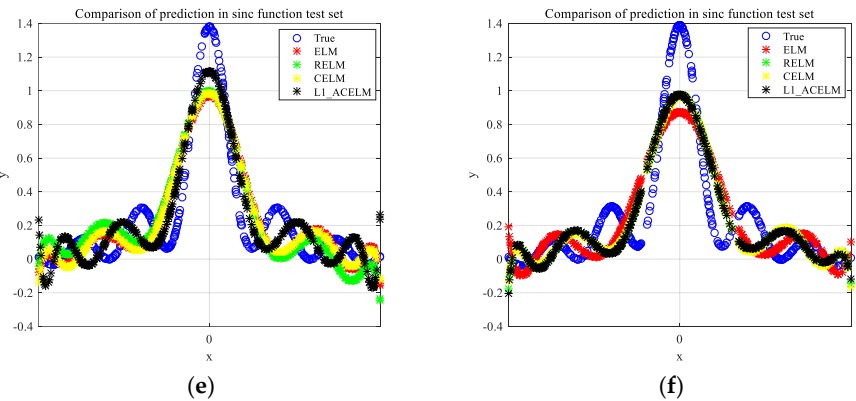

**Figure 4.** Fitting results of the sinc function with different noises.

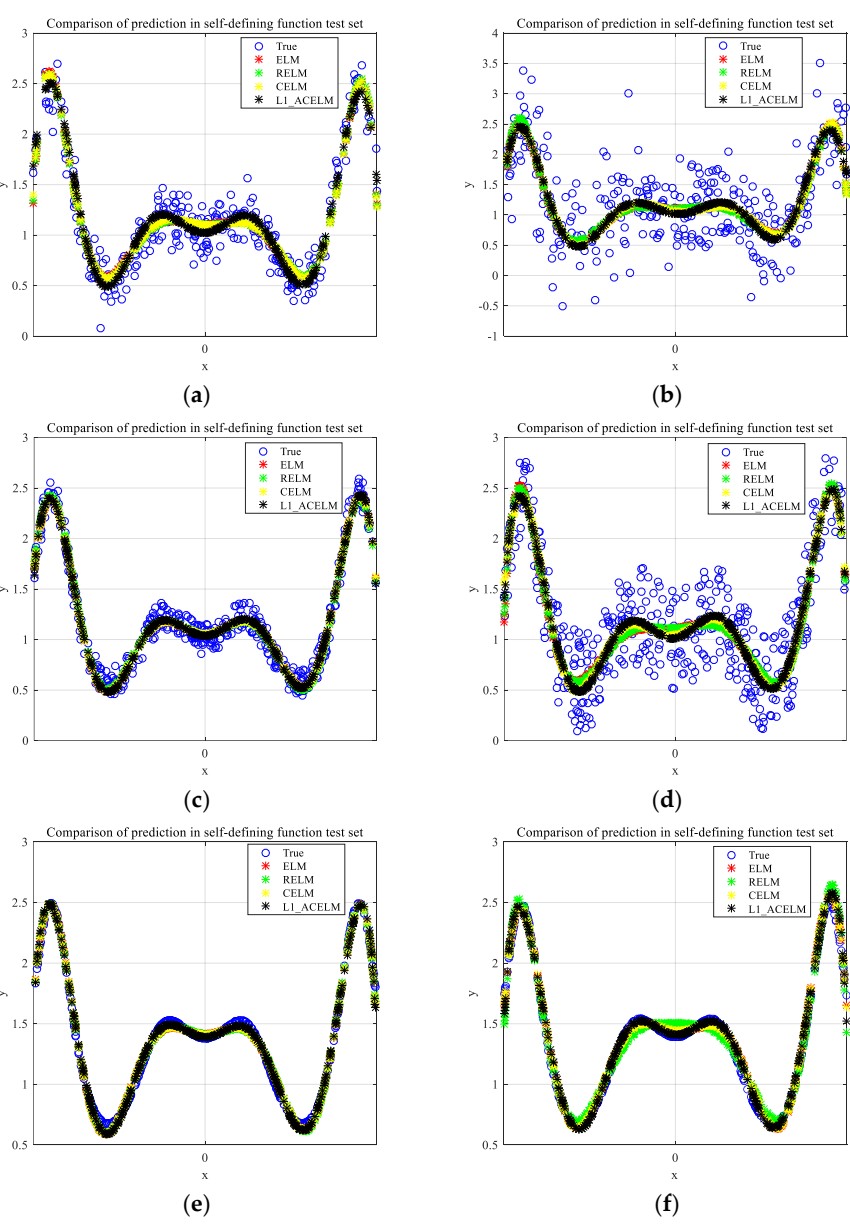

**Figure 5.** Fitting results of the self-defining function with different noises.

**Table 2.** Experiment results on artificial datasets with different types of noise.

| Dataset | Noise | Algorithm | $(\gamma,\sigma,\tau)$ | RMSE | MAE | SSE/SST | SSR/SST |
|---|---|---|---|---|---|---|---|
| | Type A | ELM | $(/,/,/)$ | 0.2429 | 0.1957 | 0.6206 | 0.3808 |
| | | RELM | $(2^{20},/,/)$ | 0.2341 | 0.1942 | 0.5768 | 0.4263 |
| | | CELM | $(2^{10},2^{-2},/)$ | 0.2345 | 0.1949 | 0.5785 | 0.4256 |
| | | $L_1-$ACELM | $(2^{-23},2^{-2},0.7)$ | **0.2109** | **0.1690** | **0.4680** | **0.5359** |
| | Type B | ELM | $(/,/,/)$ | 0.5288 | 0.4199 | 0.9064 | 0.0988 |
| | | RELM | $(2^2,/,/)$ | 0.5270 | 0.4186 | 0.9004 | 0.1004 |
| | | CELM | $(2^{-19},2^{-2},/)$ | 0.5286 | 0.4199 | 0.9060 | 0.0991 |
| | | $L_1-$ACELM | $(2^5,2^{-2},0.3)$ | **0.5221** | **0.4143** | **0.8838** | **0.1246** |
| | Type C | ELM | $(/,/,/)$ | 0.1923 | 0.1581 | 0.4332 | 0.5701 |
| | | RELM | $(2^{-42},/,/)$ | 0.2019 | 0.1677 | 0.4776 | 0.5233 |
| | | CELM | $(2^{10},2^{-2},/)$ | 0.1922 | 0.1582 | 0.4325 | 0.5705 |
| | | $L_1-$ACELM | $(2^{39},2^{-2},0.7)$ | **0.1595** | **0.1309** | **0.2978** | **0.7023** |
| | Type D | ELM | $(/,/,/)$ | 0.3262 | 0.2715 | 0.6963 | 0.7633 |
| | | RELM | $(2^{12},/,/)$ | 0.3246 | 0.2709 | 0.6890 | 0.7578 |
| | | CELM | $(2^{-38},2^{-2},/)$ | 0.3223 | 0.2695 | 0.6828 | 0.7664 |
| | | $L_1-$ACELM | $(2^{-4},2^{-2},0.3)$ | **0.3199** | **0.2678** | **0.6706** | **0.8571** |
| Sinc function | Type E | ELM | $(/,/,/)$ | 0.1737 | 0.1406 | 0.2369 | 0.7633 |
| | | RELM | $(2^{12},/,/)$ | 0.1766 | 0.1441 | 0.2451 | 0.7578 |
| | | CELM | $(2^{-12},2^{-2},/)$ | 0.1725 | 0.1398 | 0.2338 | 0.7664 |
| | | $L_1-$ACELM | $(2^{-12},2^{-2},0.2)$ | **0.1349** | **0.1175** | **0.1431** | **0.8571** |
| | Type F | ELM | $(/,/,/)$ | 0.1885 | 0.1422 | 0.2715 | 0.7298 |
| | | RELM | $(2^{-2},/,/)$ | **0.1746** | **0.1412** | **0.2328** | **0.7681** |
| | | CELM | $(2^{-1},2^{-2},/)$ | 0.1757 | 0.1413 | 0.2359 | 0.7651 |
| | | $L_1-$ACELM | $(2^{-3},2^{-2},0.1)$ | 0.1753 | 0.1416 | 0.2346 | 0.7663 |
| | Type A | ELM | $(/,/,/)$ | 0.1572 | 0.1304 | 0.0908 | 0.9105 |
| | | RELM | $(2^{-8},/,/)$ | 0.1569 | 0.1301 | 0.0893 | 0.9120 |
| | | CELM | $(2^{-7},2^{-2},/)$ | 0.1565 | 0.1294 | 0.0888 | 0.9127 |
| | | $L_1-$ACELM | $(2^{-10},2^{-2},0.5)$ | **0.1560** | **0.1241** | **0.0800** | **0.9211** |
| | Type B | ELM | $(/,/,/)$ | 0.4905 | 0.3843 | 0.4761 | 0.5251 |
| | | RELM | $(2^{26},/,/)$ | 0.4862 | 0.3850 | 0.4766 | 0.5249 |
| | | CELM | $(2^{15},2^{-2},/)$ | 0.4858 | 0.3838 | 0.4759 | 0.5252 |
| | | $L_1-$ACELM | $(2^{-16},2^{-2},0.2)$ | **0.4849** | **0.3795** | **0.4641** | **0.5369** |
| | Type C | ELM | $(/,/,/)$ | 0.0937 | 0.0794 | 0.0288 | 0.9714 |
| | | RELM | $(2^{25},/,/)$ | 0.0950 | 0.0803 | 0.0296 | 0.9706 |
| | | CELM | $(2^{17},2^{-2},/)$ | 0.0936 | 0.0792 | 0.0287 | 0.9715 |
| | | $L_1-$ACELM | $(2^{37},2^{-2},0.2)$ | **0.0934** | **0.0791** | **0.0286** | **0.9716** |
| | Type D | ELM | $(/,/,/)$ | 0.3009 | 0.2622 | 0.2471 | 0.7534 |
| | | RELM | $(2^{15},/,/)$ | 0.3006 | 0.2614 | 0.2466 | 0.7539 |
| | | CELM | $(2^{-34},2^{-2},/)$ | 0.2948 | 0.2555 | 0.2373 | 0.7634 |
| | | $L_1-$ACELM | $(2^{22},2^{-2},0.7)$ | **0.2929** | **0.2534** | **0.2342** | **0.7665** |
| | Type E | ELM | $(/,/,/)$ | 0.0434 | 0.0372 | 0.0074 | 0.9929 |
| | | RELM | $(2^{-26},/,/)$ | 0.0426 | 0.0367 | 0.0071 | 0.9932 |
| | | CELM | $(2^2,2^{-2},/)$ | 0.0425 | 0.0363 | 0.0071 | 0.9932 |
| | | $L_1-$ACELM | $(2^{44},2^{-2},0.4)$ | **0.0415** | **0.0335** | **0.0068** | **0.9935** |
| Self$-$defining function | Type F | ELM | $(/,/,/)$ | 0.0498 | 0.0425 | 0.0098 | 0.9912 |
| | | RELM | $(2^5,/,/)$ | 0.0761 | 0.0586 | 0.0230 | 0.9779 |
| | | CELM | $(2^{12},2^{-2},/)$ | **0.0481** | 0.0408 | **0.0092** | **0.9920** |
| | | $L_1-$ACELM | $(2^{20},2^{-2},0.3)$ | 0.0513 | **0.0372** | 0.0104 | 0.9908 |

Figures 4 and 5 demonstrate the fitting effect of the four algorithms on the two artificial datasets. Based on these figures, it is observed that the fitting curve of $L_1$-ACELM is the closest to the real function curve compared to the other three algorithms. In Table 2, the best test results are shown in bold.

The data in Table 2 demonstrate that $L_1$-ACELM exhibits better performance in most cases when compared to the other three algorithms for the two artificial datasets with different noises. It is evident that $L_1$-ACELM has smaller *RMSE*, *MAE*, and *SSE/SST*, and larger *SSE/SSR*. This indicates that $L_1$-ACELM is more robust to noise. For example, for the sinc function, except for F noise, the performance of the proposed algorithm is superior to that of the other algorithms for different types of noise. Moreover, it is seen that $L_1$-ACELM has better generalization performance in the case of unbalanced noise data. In conclusion, $L_1$-ACELM is more stable in a noisy environment.

### 4.3. Performance on Benchmark Datasets

To further test the robustness of $L_1$-ACELM, experiments were performed on ten UCI datasets [30] with different levels of noise, including noise-free datasets, datasets with 5% noise, and datasets with 10% noise. Noise datasets were only added to the target output value of the training datasets. Among them, datasets with 5% noise indicate that the noisy data are 5% of the training dataset. The data in the noisy dataset are randomly taken from the set $[0, d]$, where $d$ is the average of the target output values of the training datasets.

In the experiment, we randomly selected 80% of the data as the training dataset and the remaining 20% as the testing dataset for each benchmark dataset. The specific description is shown in Table 3.

**Table 3.** Description of benchmark datasets.

| Dataset | Number of Training Data | Number of Testing Data | Number of Features |
|---|---|---|---|
| Boston Housing | 404 | 102 | 13 |
| Air Quality | 7485 | 1872 | 12 |
| AutoMPG | 313 | 79 | 7 |
| Triazines | 148 | 38 | 60 |
| Bodyfat | 201 | 51 | 14 |
| Pyrim | 59 | 15 | 27 |
| Servo | 133 | 34 | 4 |
| Bike Sharing | 584 | 147 | 13 |
| Balloon | 1600 | 401 | 1 |
| $NO_2$ | 400 | 100 | 7 |

To better reflect the performance of the proposed algorithm $L_1$-ACELM, the *RMSE*, *MAE*, *SSE/SST*, and *SSR/SST* were compared with those of ELM, RELM, and CELM. The evaluation indicators and the ranking of each algorithm for different noise environments are listed in Tables 4–6, and the best test results are shown in bold. From Table 4 to Table 6, it is observed that the performance of each algorithm decreases as the noise level increases. However, compared to the other algorithms, the performance of $L_1$-ACELM is still the best in most cases. From Table 4, it can be concluded that $L_1$-ACELM performs best on nine datasets out of a total of ten datasets in term of the *RMSE* and *SSR/SST* values. Similarly, for the *MAE* and *SSE/SST* values, $L_1$-ACELM exhibits the best performance on all the datasets. Table 5 shows that after adding 5% noise, the performance of each algorithm decreases, and according to the *RMSE* value, the proposed algorithm performed well on eight of the ten datasets. For the *MAE*, *SSE/SST*, and *SSR/SST* values, $L_1$-ACELM performs better for nine datasets. Moreover, for the *RMSE*, *MAE*, and *SSR/SST* values, it exhibits superior performance in nine cases and for the *SSE/SST* values, it has better performance in all ten datasets.

**Table 4.** Performance of different algorithms under noise-free environment.

| Dataset | Algorithm | $(\gamma,\sigma,\tau)$ | RMSE | MAE | SSE/SST | SSR/SST |
|---|---|---|---|---|---|---|
| Boston Housing | ELM | $(/,/,/)$ | 4.4449(4) | 3.1736(4) | 0.2438(4) | 0.7682(4) |
| | RELM | $(2^{-16},/,/)$ | 4.1636(3) | 2.9660(2) | 0.2068(3) | 0.7998(3) |
| | CELM | $(2^{-31},2^{-2},/)$ | 4.1511(2) | 2.9847(3) | 0.2067(2) | 0.8002(2) |
| | $L_1$-ACELM | $(2^{-24},2^{-2},0.4)$ | **4.0435(1)** | **2.9236(1)** | **0.1965(1)** | **0.8097(1)** |
| Air Quality | ELM | $(/,/,/)$ | 8.3167(4) | 6.5439(4) | 0.0297(4) | 0.9705(4) |
| | RELM | $(2^{-32},/,/)$ | **7.4516(1)** | 5.7812(3) | 0.0215(2.5) | 0.9786(2) |
| | CELM | $(2^{-37},2^{-2},/)$ | 7.5140(3) | 5.7604(2) | 0.0215(2.5) | 0.9785(3) |
| | $L_1$-ACELM | $(2^{-36},2^{-2},0.4)$ | 7.4574(2) | **5.7383(1)** | **0.0212(1)** | **0.9788(1)** |
| AutoMPG | ELM | $(/,/,/)$ | 2.8296(4) | 2.0956(4) | 0.1352(4) | 0.8710(4) |
| | RELM | $(2^{-57},/,/)$ | 2.6859(3) | 1.9632(3) | 0.1205(3) | 0.8845(2) |
| | CELM | $(2^{-43},2^{-2},/)$ | 2.6590(2) | 1.9582(2) | 0.1202(2) | 0.8840(3) |
| | $L_1$-ACELM | $(2^{-32},2^{-2},0.5)$ | **2.5914(1)** | **1.8949(1)** | **0.1143(1)** | **0.8907(1)** |
| Triazines | ELM | $(/,/,/)$ | 0.0664(4) | 0.0465(4) | 0.0816(4) | 0.9283(4) |
| | RELM | $(2^{-49},/,/)$ | 0.0557(3) | 0.0410(3) | 0.0545(3) | 0.9547(3) |
| | CELM | $(2^{-19},2^{-2},/)$ | 0.0529(2) | 0.0393(2) | 0.0526(2) | 0.9573(2) |
| | $L_1$-ACELM | $(2^{-31},2^{-2},0.5)$ | **0.0490(1)** | **0.0365(1)** | **0.0416(1)** | **0.9645(1)** |
| Bodyfat | ELM | $(/,/,/)$ | 1.3123(4) | 0.7449(4) | 0.0298(4) | 0.9732(4) |
| | RELM | $(2^{-10},/,/)$ | 1.1374(3) | 0.6904(3) | 0.0233(2) | 0.9794(2) |
| | CELM | $(2^{-6},2^{-2},/)$ | 1.1352(2) | 0.6858(2) | 0.0234(3) | 0.9787(3) |
| | $L_1$-ACELM | $(2^{-16},2^{-2},0.1)$ | **1.0036(1)** | **0.5936(1)** | **0.0189(1)** | **0.9820(1)** |
| Pyrim | ELM | $(/,/,/)$ | 0.1085(4) | 0.0688(4) | 0.6897(4) | 0.6143(4) |
| | RELM | $(2^{-1},/,/)$ | 0.0759(2) | 0.0548(2) | 0.3535(2) | 0.8034(2) |
| | CELM | $(2^{-20},2^{-2},/)$ | 0.0800(3) | 0.0552(3) | 0.3839(3) | 0.7718(3) |
| | $L_1$-ACELM | $(2^{-10},2^{-2},0.1)$ | **0.0728(1)** | **0.0502(1)** | **0.2956(1)** | **0.8284(1)** |
| Servo | ELM | $(/,/,/)$ | 0.7367(4) | 0.5220(4) | 0.2826(4) | 0.7874(4) |
| | RELM | $(2^{-40},/,/)$ | 0.6769(3) | 0.4750(3) | 0.2075(3) | 0.8148(3) |
| | CELM | $(2^{-41},2^{-2},/)$ | 0.6733(2) | 0.4730(2) | 0.2061(2) | 0.8214(2) |
| | $L_1$-ACELM | $(2^{-46},2^{-2},0.4)$ | **0.6593(1)** | **0.4491(1)** | **0.1917(1)** | **0.8270(1)** |
| Bike Sharing | ELM | $(/,/,/)$ | 287.615(4) | 206.507(4) | 0.0230(4) | 0.9773(4) |
| | RELM | $(2^{-10})$ | 236.107(2) | 178.976(2) | 0.0157(2) | 0.9851(2) |
| | CELM | $(2^{-16},2^{-2},/)$ | 241.917(3) | 180.856(3) | 0.0161(3) | 0.9844(3) |
| | $L_1$-ACELM | $(2^{-9},2^{-2},0.2)$ | **217.385(1)** | **160.747(1)** | **0.0130(1)** | **0.9873(1)** |
| Balloon | ELM | $(/,/,/)$ | 0.0850(4) | 0.0543(4) | 0.3452(4) | 0.7026(4) |
| | RELM | $(2^{-29},/,/)$ | 0.0796(3) | 0.0528(3) | 0.2991(3) | 0.7147(3) |
| | CELM | $(2^{-25},2^{-2},/)$ | 0.0782(2) | 0.0527(2) | 0.2806(2) | **0.7335(1)** |
| | $L_1$-ACELM | $(2^{-24},2^{-2},0.9)$ | **0.0773(1)** | **0.0525(1)** | **0.2790(1)** | 0.7304(2) |
| NO$_2$ | ELM | $(/,/,/)$ | 0.5272(4) | 0.4128(4) | 0.5157(4) | 0.5060(4) |
| | RELM | $(2^{-9},/,/)$ | 0.5154(2) | 0.4034(2) | 0.4844(2) | 0.5298(2) |
| | CELM | $(2^{-15},2^{-2},/)$ | 0.5161(3) | 0.4047(3) | 0.4910(3) | 0.5271(3) |
| | $L_1$-ACELM | $(2^{-17},2^{-2},0.2)$ | **0.5132(1)** | **0.4028(1)** | **0.4823(1)** | **0.5338(1)** |

**Table 5.** Performance of different algorithms under 5% noise environment.

| Dataset | Algorithm | $(\gamma, \sigma, \tau)$ | RMSE | MAE | SSE/SST | SSR/SST |
|---|---|---|---|---|---|---|
| Boston Housing | ELM | $(/, /, /)$ | 6.5817(4) | 4.1292(4) | 0.4196(4) | 0.5962(4) |
| | RELM | $(2^{-17}, /, /)$ | 6.2972(3) | 3.9095(3) | 0.3835(3) | 0.6327(3) |
| | CELM | $(2^{-6}, 2^{-2}, /)$ | 6.2155(2) | 3.8937(2) | 0.3756(2) | 0.6407(2) |
| | $L_1$-ACELM | $(2^{-5}, 2^{-2}, 0.5)$ | **6.1256**(1) | **3.8185**(1) | **0.3675**(1) | **0.6478**(1) |
| Air Quality | ELM | $(/, /, /)$ | 12.0381(4) | 7.5222(4) | 0.0531(4) | 0.9471(4) |
| | RELM | $(2^{-32}, /, /)$ | 11.6199(2) | 7.1866(3) | 0.0496(2) | 0.9504(2) |
| | CELM | $(2^{-39}, 2^{-2}, /)$ | 11.6303(3) | 7.1554(2) | 0.0499(3) | 0.9501(3) |
| | $L_1$-ACELM | $(2^{-39}, 2^{-2}, 0.8)$ | **11.5540**(1) | **7.1145**(1) | **0.0489**(1) | **0.9511**(1) |
| AutoMPG | ELM | $(/, /, /)$ | 5.6949(4) | 3.2315(4) | 0.4024(4) | 0.6204(4) |
| | RELM | $(2^{-21}, /, /)$ | 5.5923(2) | 3.1677(3) | 0.3919(3) | 0.6337(2) |
| | CELM | $(2^{-28}, 2^{-2}, /)$ | 5.6502(3) | 3.1189(2) | 0.3915(2) | 0.6299(3) |
| | $L_1$-ACELM | $(2^{-30}, 2^{-2}, 0.9)$ | **5.4775**(1) | **3.0347**(1) | **0.3688**(1) | **0.6558**(1) |
| Triazines | ELM | $(/, /, /)$ | 0.0937(4) | 0.0618(4) | 0.1510(4) | 0.8719(4) |
| | RELM | $(2^{-16}, /, /)$ | 0.0790(3) | 0.0549(3) | 0.1031(3) | 0.9199(3) |
| | CELM | $(2^{-39}, 2^{-2}, /)$ | 0.0779(2) | 0.0515(2) | 0.0989(2) | 0.9172(2) |
| | $L_1$-ACELM | $(2^{-22}, 2^{-2}, 0.5)$ | **0.0725**(1) | **0.0489**(1) | **0.0834**(1) | **0.9273**(1) |
| Bodyfat | ELM | $(/, /, /)$ | 4.1325(4) | 2.0890(4) | 0.2414(4) | 0.7783(4) |
| | RELM | $(2^{-16}, /, /)$ | 3.9255(3) | 2.0575(3) | 0.2115(3) | 0.8027(2) |
| | CELM | $(2^{-36}, 2^{-2}, /)$ | 3.8868(2) | 2.0413(2) | 0.2095(2) | 0.8078(3) |
| | $L_1$-ACELM | $(2^{-11}, 2^{-2}, 0.6)$ | **3.7288**(1) | **1.9119**(1) | **0.1986**(1) | **0.8149**(1) |
| Pyrim | ELM | $(/, /, /)$ | 0.1019(4) | 0.0722(4) | 0.6711(4) | 0.6685(4) |
| | RELM | $(2^{-12}, /, /)$ | 0.0825(2) | 0.0591(2) | 0.4008(2) | 0.7537(2) |
| | CELM | $(2^{-3}, 2^{-2}, /)$ | 0.0871(3) | 0.0609(3) | 0.4435(3) | 0.7153(3) |
| | $L_1$-ACELM | $(2^{-13}, 2^{-2}, 0.8)$ | **0.0743**(1) | **0.0562**(1) | **0.3720**(1) | **0.7762**(1) |
| Servo | ELM | $(/, /, /)$ | 0.8424(4) | 0.5868(4) | 0.3224(4) | 0.7235(4) |
| | RELM | $(2^{-46}, /, /)$ | 0.7753(3) | 0.5473(3) | 0.2794(3) | 0.7742(3) |
| | CELM | $(2^{-42}, 2^{-2}, /)$ | **0.7598**(1) | **0.5252**(1) | **0.2763**(1) | 0.7752(2) |
| | $L_1$-ACELM | $(2^{-49}, 2^{-2}, 0.7)$ | 0.7724(2) | 0.5299(2) | 0.2983(2) | **0.7778**(1) |
| Bike Sharing | ELM | $(/, /, /)$ | 1130.04(4) | 497.051(4) | 0.2730(4) | 0.7352(4) |
| | RELM | $(2^{-1}, /, /)$ | 1093.85(2) | 453.720(2) | 0.2556(3) | 0.7505(3) |
| | CELM | $(2^{-9}, 2^{-2}, /)$ | 1094.35(3) | 461.094(3) | 0.2545(2) | **0.7523**(1.5) |
| | $L_1$-ACELM | $(2^{-6}, 2^{-2}, 0.9)$ | **1085.27**(1) | **441.646**(1) | **0.2526**(1) | **0.7523**(1.5) |
| Balloon | ELM | $(/, /, /)$ | 0.0874(4) | 0.0546(3) | 0.3815(4) | 0.6794(4) |
| | RELM | $(2^{-16}, /, /)$ | 0.0850(3) | 0.0544(2) | 0.3444(3) | 0.7170(2) |
| | CELM | $(2^{-9}, 2^{-2}, /)$ | 0.0799(2) | 0.0549(4) | 0.3086(2) | 0.7135(3) |
| | $L_1$-ACELM | $(2^{-5}, 2^{-2}, 0.9)$ | **0.0782**(1) | **0.0536**(1) | **0.2704**(1) | **0.7368**(1) |
| NO$_2$ | ELM | $(/, /, /)$ | **0.9489**(1) | 0.5767(2) | 0.7594(2) | **0.2803**(1) |
| | RELM | $(2^{-31}, /, /)$ | 0.9698(3) | 0.5781(3) | 0.7754(3) | 0.2692(3) |
| | CELM | $(2^{-19}, 2^{-2}, /)$ | 0.9737(4) | 0.5856(4) | 0.7844(4) | 0.2644(4) |
| | $L_1$-ACELM | $(2^{-19}, 2^{-2}, 0.5)$ | 0.9611(2) | **0.5708**(1) | **0.7515**(1) | 0.2790(2) |

**Table 6.** Performance of different algorithms under 10% noise environment.

| Dataset | Algorithm | $(\gamma, \sigma, \tau)$ | RMSE | MAE | SSE/SST | SSR/SST |
|---|---|---|---|---|---|---|
| Boston Housing | ELM | $(/, /, /)$ | 8.6315(4) | 5.1524(4) | 0.5873(4) | 0.4557(4) |
| | RELM | $(2^{-30}, /, /)$ | 8.2456(3) | 5.1512(3) | 0.5177(3) | 0.4999(3) |
| | CELM | $(2^{-36}, 2^{-2}, /)$ | 8.2437(2) | 4.9250(2) | 0.5151(2) | 0.5006(2) |
| | $L_1$-ACELM | $(2^{-48}, 2^{-2}, 0.9)$ | **8.1718**(1) | **4.8090**(1) | **0.5123**(1) | **0.5074**(1) |
| Air Quality | ELM | $(/, /, /)$ | 14.7386(4) | 8.8277(4) | 0.0778(4) | 0.9223(4) |
| | RELM | $(2^{-39}, /, /)$ | 14.5651(3) | 8.4928(3) | 0.0759(3) | 0.9241(3) |
| | CELM | $(2^{-45}, 2^{-2}, /)$ | 14.5412(2) | 8.4737(2) | 0.0754(2) | 0.9246(2) |
| | $L_1$-ACELM | $(2^{-4}, 2^{-2}, 0.6)$ | **14.4355**(1) | **8.4236**(1) | **0.0748**(1) | **0.9253**(1) |
| AutoMPG | ELM | $(/, /, /)$ | 7.0139(3) | 4.0307(2) | 0.5218(3) | 0.5009(4) |
| | RELM | $(2^{-28}, /, /)$ | 7.0729(4) | 4.0592(3) | 0.5278(4) | 0.5068(3) |
| | CELM | $(2^{-27}, 2^{-2}, /)$ | 6.9306(2) | 4.0792(4) | 0.5147(2) | **0.5183**(1) |
| | $L_1$-ACELM | $(2^{-39}, 2^{-2}, 0.1)$ | **6.9151**(1) | **3.9845**(1) | **0.5032**(1) | 0.5169(2) |
| Triazines | ELM | $(/, /, /)$ | 0.1166(4) | 0.0776(4) | 0.2077(4) | 0.8116(4) |
| | RELM | $(2^{-37}, /, /)$ | 0.1068(2) | 0.0703(2) | 0.1693(2) | 0.8536(2) |
| | CELM | $(2^{-21}, 2^{-2}, /)$ | 0.1074(3) | 0.0705(3) | 0.1729(3) | 0.8501(3) |
| | $L_1$-ACELM | $(2^{-29}, 2^{-2}, 0.6)$ | **0.0963**(1) | **0.0638**(1) | **0.1378**(1) | **0.8815**(1) |
| Bodyfat | ELM | $(/, /, /)$ | 6.5116(3) | 3.4749(2) | 0.4184(4) | 0.6129(4) |
| | RELM | $(2^{-23}, /, /)$ | 6.5075(2) | 3.4977(3) | 0.4094(2) | 0.6180(3) |
| | CELM | $(2^{-22}, 2^{-2}, /)$ | 6.5343(4) | 3.5697(4) | 0.4119(3) | 0.6182(2) |
| | $L_1$-ACELM | $(2^{-8}, 2^{-2}, 0.4)$ | **6.3088**(1) | **3.4931**(1) | **0.3743**(1) | **0.6515**(1) |
| Pyrim | ELM | $(/, /, /)$ | 0.1263(4) | 0.0903(4) | 0.9389(4) | 0.5540(4) |
| | RELM | $(2^{-23}, /, /)$ | 0.1136(2) | 0.0804(2) | 0.7002(2) | 0.6048(3) |
| | CELM | $(2^{-10}, 2^{-2}, /)$ | 0.1137(3) | 0.0812(3) | 0.7098(3) | 0.6515(2) |
| | $L_1$-ACELM | $(2^{-24}, 2^{-2}, 0.5)$ | **0.1010**(1) | **0.0717**(1) | **0.4848**(1) | **0.7080**(1) |
| Servo | ELM | $(/, /, /)$ | 0.8648(4) | 0.6291(3) | 0.3719(4) | 0.7042(4) |
| | RELM | $(2^{-34}, /, /)$ | 0.8253(3) | 0.6889(4) | 0.2863(3) | 0.7633(2) |
| | CELM | $(2^{-39}, 2^{-2}, /)$ | 0.8025(2) | 0.5487(2) | 0.2788(2) | 0.7557(3) |
| | $L_1$-ACELM | $(2^{-45}, 2^{-2}, 0.9)$ | **0.7486**(1) | **0.5332**(1) | **0.2412**(1) | **0.7960**(1) |
| Bike Sharing | ELM | $(/, /, /)$ | 1614.52(4) | 755.097(4) | 0.4224(4) | 0.5926(4) |
| | RELM | $(2^{-39}, /, /)$ | 1587.01(3) | 716.147(2) | 0.4052(3) | 0.6055(3) |
| | CELM | $(2^{-42}, 2^{-2}, /)$ | 1582.54(2) | 718.328(3) | 0.4012(2) | 0.6089(2) |
| | $L_1$-ACELM | $(2^{-49}, 2^{-2}, 0.1)$ | **1562.74**(1) | **714.710**(1) | **0.3952**(1) | **0.6194**(1) |
| Balloon | ELM | $(/, /, /)$ | **0.0785**(1) | 0.0547(3) | 0.2749(2) | 0.7321(2) |
| | RELM | $(2^{-34}, /, /)$ | 0.0807(4) | 0.0549(4) | 0.2871(3) | 0.7206(3) |
| | CELM | $(2^{-39}, 2^{-2}, /)$ | 0.0793(3) | 0.0545(2) | 0.2931(4) | 0.7127(4) |
| | $L_1$-ACELM | $(2^{-42}, 2^{-2}, 0.5)$ | 0.0788(2) | **0.0544**(1) | **0.2682**(1) | **0.7398**(1) |
| NO$_2$ | ELM | $(/, /, /)$ | 1.2576(4) | **0.7013**(1) | 0.8752(3) | 0.1643(4) |
| | RELM | $(2^{-16}, /, /)$ | 1.2718(2) | 0.7259(4) | 0.8908(4) | 0.1663(3) |
| | CELM | $(2^{-27}, 2^{-2}, /)$ | 1.2478(3) | 0.7164(3) | 0.8639(2) | 0.1770(2) |
| | $L_1$-ACELM | $(2^{-23}, 2^{-2}, 0.2)$ | **1.2408**(1) | 0.7080(2) | **0.8566**(1) | **0.1882**(1) |

To further illustrate the difference between the proposed algorithm and traditional algorithms, we conducted statistical analysis on the experimental results. Friedman's test [31] is a well-known test for comparing the performance of various algorithms on datasets. Tables 7–9 list the average ranks of four algorithms on four performance measures under a noise-free environment and noisy environment.

**Table 7.** Average ranks of benchmark algorithms under noise-free environment.

| Algorithm | RMSE | MAE | SSE/SST | SSR/SST |
|-----------|------|-----|---------|---------|
| ELM | 4 | 4 | 4 | 4 |
| RELM | 2.5 | 2.6 | 2.55 | 2.4 |
| CELM | 2.4 | 2.4 | 2.45 | 2.5 |
| $L_1$-ACELM | 1.1 | 1.0 | 1.0 | 1.1 |

**Table 8.** Average ranks of benchmark algorithms under 5% noise environment.

| Algorithm | RMSE | MAE | SSE/SST | SSR/SST |
|-----------|------|-----|---------|---------|
| ELM | 3.7 | 3.7 | 3.8 | 3.7 |
| RELM | 2.6 | 2.7 | 2.8 | 2.5 |
| CELM | 2.5 | 2.5 | 2.3 | 2.65 |
| $L_1$-ACELM | 1.0 | 1.1 | 1.1 | 1.15 |

**Table 9.** Average ranks of benchmark algorithms under 10% noise environment.

| Algorithm | RMSE | MAE | SSE/SST | SSR/SST |
|-----------|------|-----|---------|---------|
| ELM | 3.5 | 3.1 | 3.6 | 3.8 |
| RELM | 2.8 | 3.0 | 2.9 | 2.8 |
| CELM | 2.6 | 2.8 | 2.5 | 2.3 |
| $L_1$-ACELM | 1.1 | 1.1 | 1.0 | 1.1 |

The Friedman statistic variable can be expressed as follows:

$$\chi_F^2 = \frac{12N}{k(k+1)} \left[ \sum_j R_j^2 - \frac{k(k+1)^2}{4} \right] \tag{44}$$

which is distributed according to $\chi_F^2$ with $k-1$ degrees of freedom, where $R_j$ is the average rank of the algorithms as listed in Tables 7–9. $N = 10$ and $k = 4$ are the number of datasets and the number of the algorithms, respectively. The Friedman statistic follows an F-distribution:

$$F_F = \frac{(N-1)\chi_F^2}{N(k-1) - \chi_F^2} \tag{45}$$

with $k-1$ and $(k-1)(N-1)$ degrees of freedom. Table 10 shows the results of the Friedman test on the dataset without noise, with 5% noise, and with 10% noise. For $\alpha = 0.05$, the critical value of $F_\alpha(3,27)$ is 2.960. For the four algorithms, ELM, RELM, CELM, and $L_1$-ACELM, $F_F > F_\alpha$ is achieved by comparing the results from Table 10. Therefore, the assumption that all the algorithms perform the same is rejected. To further contrast the differences between paired algorithms, the Nemenyi test [32] is often used as a post hoc test.

**Table 10.** Relevant values in the Friedman test on benchmark datasets.

| Ratio of Noise | $\chi_F^2$ | | | | $F_F$ | | | | CD |
|----------------|------|-----|---------|---------|------|-----|---------|---------|----|
| | RMSE | MAE | SSE/SST | SSR/SST | RMSE | MAE | SSE/SST | SSR/SST | |
| Noise-free | 25.32 | 27.12 | 27.03 | 25.32 | 48.69 | 84.75 | 81.91 | 48.69 | 1.4832 |
| 5% noise | 16.20 | 20.64 | 22.68 | 19.71 | 10.57 | 19.81 | 27.89 | 17.24 | 1.4832 |
| 10% noise | 18.36 | 15.96 | 21.72 | 22.68 | 14.20 | 10.23 | 23.61 | 27.89 | 1.4832 |

The critical difference can be expressed as:

$$CD = q_\alpha \sqrt{\frac{k(k+1)}{6N}} = 2.569 \times \sqrt{\frac{4 \times (4+1)}{6 \times 10}} = 1.4832 \tag{46}$$

where the critical value of $q_{0.05}$ is 2.569. Here, we can compare the average rank difference between the proposed algorithm and other algorithms using the *CD* value. If the average rank difference is greater than the *CD* value, this implies that the proposed algorithm is superior to the other algorithms. Otherwise, there is no difference between the two algorithms. Therefore, we can analyze the difference between the proposed algorithm and other algorithms in the following three cases:

(1) Under noise-free environment. For the *RMSE* and *SSR/SST* index, the performance of $L_1$-ACELM is better than that of ELM ($4 - 1.1 = 2.9 > 1.4832$). For the *MAE* index, the performance of $L_1$-ACELM is better than that of ELM ($4 - 1.0 = 3.0 > 1.4832$) and RELM ($2.6 - 1.0 = 1.5 > 1.4832$). There is no significant difference between $L_1$-ACELM and CELM.

(2) Under 5% noise environment. For the *RMSE* index, the performance of $L_1$-ACELM is better than that of ELM ($3.7 - 1.0 = 2.7 > 1.4832$), RELM ($2.6 - 1.0 = 1.6 > 1.4832$), and CELM ($2.5 - 1.0 = 1.5 > 1.4832$). For the *MAE* and *SSE/SST* index, the performance of $L_1$-ACELM is better than that of ELM ($3.7 - 1.1 = 2.6 > 1.4832$, $3.8 - 1.1 = 2.7 > 1.4832$) and RELM ($2.7 - 1.1 = 1.6 > 1.4832$, $2.8 - 1.1 = 1.7 > 1.4832$). For the *SSR/SST* index, the performance of $L_1$-ACELM is better than that of ELM ($3.7 - 1.15 = 2.55 > 1.4832$) and CELM ($2.65 - 1.15 = 1.5 > 1.4832$).

(3) Under 10% noise environment. Similarly, for the *RMSE*, *MAE*, and *SSE/SST* index, the performance of $L_1$-ACELM is better than that of ELM, RELM, and CELM. For the *SSR/SST* index, the performance of $L_1$-ACELM is better than that of ELM and RELM.

## 5. Conclusions

In this paper, a novel asymmetric, bounded, smooth non-convex loss function based on the expected loss and the correntropy loss is proposed, termed AC-loss. The AC-loss loss function and $L_1$ norm are introduced into the regularized extreme learning machine, and an improved robust regularized extreme learning machine is proposed for regression. Owing to the non-convexity of the AC-loss function, it is difficult to solve $L_1$-ACELM. As such, the half-quadratic optimization algorithm is applied to address the nonconvex optimization problem. To prove the effectiveness of $L_1$-ACELM, experiments are conducted on artificial datasets and benchmark datasets with different types of noise, respectively. The results demonstrate the significant advantages of $L_1$-ACELM in generalization performance and robustness, especially when the data distribution with noise and outliers are asymmetric.

The PGD algorithm is used to solve the $L_1$-ACELM in this paper. Since it is an iterative process, the training speed is reduced. In the future, we will research a faster method to solve this optimization problem.

**Author Contributions:** Conceptualization, Q.W. and F.W.; methodology, Q.W.; software, F.W.; validation, F.W., Y.A. and K.L.; writing—original draft preparation, F.W.; writing—review and editing, Q.W.; visualization, Y.A.; funding acquisition, Q.W. All authors have read and agreed to the published version of the manuscript.

**Funding:** This research was funded by the National Natural Science Foundation of China under Grant (51875457), the Key Research Project of Shaanxi Province (2022GY-050, 2022GY-028), the Natural Science Foundation of Shaanxi Province of China (2022JQ-636, 2021JQ-701, 2021JQ-714), and Shaanxi Youth Talent Lifting Plan of Shaanxi Association for Science and Technology (20220129).

**Data Availability Statement:** The data presented in the article are freely available and are listed at the reference address in the bibliography.

**Conflicts of Interest:** The authors declare no conflict of interest.

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
