# Peer review of "L1-Norm Robust Regularized Extreme Learning Machine with Asymmetric C-Loss for Regression"

_axioms, doi:10.3390/axioms12020204_

Round 1
Reviewer 1 Report
Revision Report
In this work, the study field of machine learning/artificial intelligence is focused, involving the subject of Extreme Learning Machines (ELM). The manuscript proposes a loss function (called AC-loss) to be used in ELM. Based on this function, a new ELM is proposed (called L1-ACELM) which uses robust regularization based on L1 norm.
The paper is overall well written and well explained. The research is well conducted. However, there are some few questions that appeared in the reading of the manuscript. See the report below.
Issues:
- The developed code and used datasets (the artificial ones) should be made available publicly (e.g., in a public repository such as GitHub, Bitbucket, or other repository, used to publish data and code associated with an article, for other readers to test and reproduce the article's results. Suitable links should be provided in the manuscript.
- In the conclusions, the authors mention execution times issues with the algorithm PGD. In the experimental results, execution times should be included for the tested algorithms.
Minor issues:
p. 1, Sect. 1:
"ELM based" should be "ELM is based"
"often leads" should be "which often leads"
p. 2: "In OP-ELM, the L 1 norm is used for the output nodes is used to" The sentence does not read well, please improve the English language.
p. 5: The set of real numbers symbol should be represented by the Latin capital letter “R” presented with a double struck typeface.
p. 8: "it is possible to the optimal solution" The sentence does not read well, please improve the sentence writing.
p. 9: "respectively. we ..." Writing suggestion: "... respectively. We .... and the pseudo code is presented in Algorithm 1."
p. 11: "a means of zero" should be "a mean of zero"
p. 16: "algorithm was" In English writing, one typically uses the present tense when describing results. Suggestion: "algorithm is".
p. 17:
Third column of Table 3: Do you mean "The number of testing data"?
"From Table 4Table 6," should be "From Table 4 to Table 6"
p. 20:
"was better" should be "is better" (replace in other places as well)
"There was" should be "There is"
Author Response
We are very grateful for your critical comments and constructive suggestions and for giving us the opportunity to revise our manuscript. We have carefully revised the manuscript according to your valuable suggestions. Please see the attachment for detailed amendments.

Reviewer 2 Report
accept in current form
Author Response

(The authors gave the same response as above.)

Reviewer 3 Report
The submitted manuscript (ms) proposes alternative loss functions for extreme learning machines (ELM). In general, I found the topic novel and interesting. The ms could improve with more careful language, table, and figure editing.
Major comments:
1. Abstract, line 12: write “asymmetric, non-convex and bounded …”. These adjectives are also used later in the text.
2. Abstract, line 13: why use the acronym “AC-loss”? Or is the motivation explained later in the text?
3. Section 2 and others: Carefully look for mathematical symbols within the text. They should also appear in italic font (e.g., “L”, “j”).
4. Equation 3: which norm is used? I suppose it is the euclidean norm?
5. 206: use a new equation for defining u.
6. Table 2: Include labels for the respective quantities (“RMSE”, “MAE”, …) in each corresponding row in the table.
Minor comments:
7. 66: language issue “Quantile can fully …”
8. 102: language issue “leads to difficulty …”
9. 107: write “we propose …”
10. 109: write “we analyze …”
11. 182: write “least square loss …”
Author Response

(The authors gave the same response as above.)
